# Therapeutic Potential of IL-15 and N-803 in HIV/SIV Infection

**DOI:** 10.3390/v13091750

**Published:** 2021-09-02

**Authors:** Olivia Harwood, Shelby O’Connor

**Affiliations:** Department of Pathology and Laboratory Medicine, University of Wisconsin-Madison, Madison, WI 53711, USA; osayer@wisc.edu

**Keywords:** IL-15, N-803, SIV, HIV

## Abstract

IL-15, a proinflammatory cytokine critical for the generation, maintenance, and homeostasis of T cell responses, is produced naturally in response to HIV/SIV infection, but has also demonstrated therapeutic potential. IL-15 can boost CD4^+^ and CD8^+^ T cell and NK cell proliferation, activation, and function. However, IL-15 treatment may cause aberrant immune activation and accelerated disease progression in certain circumstances. Moreover, the relationship between the timing of IL-15 administration and disease progression remains unclear. The IL-15 superagonist N-803 was developed to expand the therapeutic potential of IL-15 by maximizing its tissue distribution and half-life. N-803 has garnered enthusiasm recently as a way to enhance the innate and cellular immune responses to HIV/SIV by improving CD8^+^ T cell recognition and killing of virus-infected cells and directing immune cells to mucosal sites and lymph nodes, the primary sites of virus replication. N-803 has also been evaluated in “shock and kill” strategies due to its potential to reverse latency (shock) and enhance antiviral immunity (kill). This review examines the current literature about the effects of IL-15 and N-803 on innate and cellular immunity, viral burden, and latency reversal in the context of HIV/SIV, and their therapeutic potential both alone and combined with additional interventions such as antiretroviral therapy (ART) and vaccination.

## 1. Introduction

Cytotoxic CD8^+^ T cells play a crucial role in responding to viral infections such as human immunodeficiency virus (HIV). HIV preferentially infects CD4^+^ T cells, as well as macrophages and monocytes, in which a latent reservoir is established [1]. Infection with HIV also leads to the death of both infected and uninfected helper CD4^+^ T cells [2]. CD8^+^ T cells naturally mount a robust immune response to HIV infection to detect and kill infected cells, and secrete cytokines and chemokines, among other functions. Despite this, the immune response fails to fully suppress virus replication in the majority of HIV-infected individuals [3,4]. If left untreated, HIV-infected individuals progress to acquired immunodeficiency syndrome (AIDS), which is characterized by increasing plasma viremia and corresponding depletion of CD4^+^ T cells through complex immune dysfunction including CD4^+^ T cell death and CD8^+^ T cell activation and expansion [5,6,7,8].

Antiretroviral therapy (ART) may allow for CD4^+^ T cell reconstitution, but this process is slow and dependent upon the CD4^+^ T cell count prior to ART initiation [9,10]. The success of ART also depends upon adherence, but even those who are durably suppressed on ART may experience immune dysregulation [11]. While ART can suppress virus replication and help normalize T cell features, such as activation status and CD4/CD8 ratio, disrupted by HIV/SIV infection, ART is not a cure [12,13]. If ART is discontinued, HIV replication can resume from the latently infected cells, leading to viral rebound, and the immune system typically cannot control virus replication on its own [11,14]. For these reasons, additional therapeutic options, including immunotherapy, are being explored to bolster the immune response to HIV to achieve a “functional cure” in which the immune system is able to control HIV, rendering ART unnecessary [2,15].

One such strategy to generate a “functional cure” includes modulating the expression of certain proinflammatory cytokines to improve immune competence and boost the anti-HIV immune response to better control HIV replication, independent of ongoing ART [16,17]. The common gamma chain cytokines, including interleukin-15 (IL-15), have been a major focus of interest because they help regulate immune cell development, maintenance, differentiation, proliferation, and function. In this review, we summarize the current research on free IL-15 and N-803, an IL-15 superagonist, in the context of therapeutic interventions for treated and untreated HIV infection in humans and SIV (simian immunodeficiency virus) infection in nonhuman primates.

## 2. IL-15 Signaling, Function, and Immunotherapeutic Potential

IL-15 is a cytokine produced by mononuclear phagocytes, dendritic cells, and T cells. It is a critical factor for the development, proliferation, maintenance, and activation of T cells and natural killer (NK) cells [18]. Specifically, it has been shown to boost memory and antigen-specific CD8^+^ T cells, making it attractive for therapeutic use to stimulate the cellular immune response against various diseases [19]. IL-15 transcripts are expressed abundantly in a wide variety of tissues, such as skeletal muscle, kidney, and lung, and cell types, including epithelial cells, monocytes, macrophages, and dendritic cells [20,21]. IL-15 protein expression is heavily regulated by multiple negative feedback loops to prevent aberrant immune activation and autoimmunity. IL-15 signaling induced by viral infections is vital in converting naïve immune cells, particularly T cells and NK cells, into effector cells capable of expansion and ultimately pathogen recognition and elimination [20].

Type I interferons (IFNs, e.g., IFN-α), among other factors, are also produced rapidly in response to infections such as HIV/SIV, and they can induce production of IL-15 [22]. While Type I IFNs are typically thought of as innate immune cytokines, they also mediate facets of adaptive immunity, including IL-15 signaling [8,23]. Type I IFN signaling has been shown to increase both the expression of the IL-15Rα subunit, as well as the cytotoxicity of T cells and NK cells through IL-15 signaling [23]. In acute infection, Type I IFN signaling promotes beneficial innate and adaptive immune responses, including IL-15 signaling. However, excessive Type I IFN signaling contributes to aberrant immune activation, worsening HIV disease burden, and poor CD4^+^ T cell reconstitution under ART during chronic HIV infection [24,25]. IL-15 signaling, as well as the factors such as Type I IFNs that induce IL-15 production, expression, and signaling, must therefore be carefully evaluated for the ability to promote both beneficial and detrimental immunomodulatory effects throughout acute and chronic HIV/SIV.

IL-15 shares some features with interleukin-2 (IL-2), but has distinct functions. The common gamma chain cytokine IL-2 has been studied extensively in HIV because it can activate immune cells and prevent their apoptosis. Administration of IL-2 can also cause substantial toxicity, so there is growing interest in cytokines that may be modulated to augment immune function with less toxicity. IL-15, which is critical for mediating the adaptive immune response, is one of at least six cytokines that utilize the common gamma chain (ɣc) receptor subunit, also known as the IL-2 receptor gamma subunit (IL-2Rɣ, CD132) [26,27,28]. In addition to the ɣc receptor subunit, IL-15 also shares the β subunit of the IL-2 receptor (IL-2Rβ, CD122) with IL-2, making the signaling complex formed by the heterodimeric β and ɣc receptor subunits nearly identical for both IL-2 and IL-15 [29]. One distinguishing feature between IL-2 and IL-15 is that each cytokine utilizes a third receptor unit, termed IL-2Rα (CD25) and IL-15Rα (CD215), respectively. The α subunits are cytokine-specific, and together form the high-affinity forms of IL-2R and IL-15R [29]. Unlike IL-2, IL-15 signaling is uniquely mediated by the IL-15 cytokine bound to the IL-15Rα subunit. The IL-15/IL-15Rα complex is then presented *in cis* to the IL-15Rβɣc complex, or *in trans* to neighboring cells bearing the IL-15Rβɣc complex [28,30]. IL-2Rα alone exhibits low affinity for IL-2 in the absence of the IL-2Rβɣc. Free IL-2 can effectuate signaling by binding with the complete IL-2R complex (IL-2Rαβɣc) [31]. However, IL-15Rα binds IL-15 with high affinity in the absence of the IL-2Rβɣc, thereby enabling the presentation of the IL-15/IL-15Rα complex to the IL-15Rβɣc complex *in cis* or *in trans*.

IL-2 and IL-15 have similar 3D structures and they share some signaling pathways. IL-2 and IL-15 are both members of the ‘four α-helix bundle’ family [21,32], even though their primary sequences are quite different. IL-2 and IL-15 also signal specifically through JAK1/3 and STAT3/5 to induce cellular activation [27,33,34]. They also stimulate the proliferation of activated T cells and facilitate the generation and proliferation of antigen-specific effector T cells and the proliferation and immunoglobulin synthesis by B cells [35]. However, IL-2 and IL-15 also have opposing functions. IL-2 signaling generally activates regulatory T (Treg) cells and mediates activation-induced cell death, both of which are critical to preventing autoimmunity, yet are not ideal features of HIV immunotherapy [36,37]. On the other hand, IL-15 signaling is critical for maintaining memory T cell responses to pathogens, which is a vital aspect of HIV immunotherapy [18].

There are numerous scenarios in which these distinctive features of IL-15 signaling may be agonized or antagonized to promote immunomodulation. For example, elevated IL-15 signaling relative to baseline has been associated with pathogenesis of autoimmune disorders, so in that context, it may be beneficial to inhibit IL-15 signaling [38]. Conversely, IL-15 signaling may be critical in helping mount an effective antiviral immune response capable of mediating viral control in HIV/SIV. Thus, IL-15 is an attractive candidate target for immunotherapy in chronic diseases such as HIV and SIV. Since it can maintain, expand, and stimulate antigen-specific CD8^+^ T cells and NK cells, IL-15 could be used in the context of HIV to enable the immune system to target and eliminate infected cells, thereby improving the lives of people living with HIV. However, free IL-15 has drawbacks such as a short half-life and limited biological activity *in vivo*. As a result, molecules to agonize IL-15 signaling have been developed to overcome some of the limitations of free IL-15. Both free IL-15 and IL-15 agonist complexes are being studied preclinically and clinically for their immunotherapeutic potential.

## 3. IL-15 Dosing and Toxicity in Macaques

Many groups have evaluated the safety, toxicity, dosing, and effects of IL-15 treatment on macaques. Human and macaque free IL-15 are generally produced in an E. coli or mammalian cell line expression system [39,40]. Studies in macaques have explored alternative doses (2.5–100 μg/kg), frequencies (continuous, daily, or intermittent), and routes (intravenous and subcutaneous) for the administration of free IL-15, as well as the ability of free IL-15 to bind soluble or membrane bound IL-15Rα to facilitate cellular immunomodulation [30,41,42,43,44,45].

Daily and intermittent (2–7 days between doses) subcutaneous IL-15 administration strategies were evaluated in pigtail and rhesus macaques [30,44]. Daily administration of IL-15 induced distinct, yet reversible, toxicity measured by weight loss and a transient decrease (to 90/μL) in absolute neutrophil count (ANC) that returned to normal levels (~2000/μL) upon treatment cessation [30]. Intermittent administration of IL-15 resulted in similar peak plasma IL-15 levels to daily administration, but no toxicity was observed, likely due to cytokine clearance between doses [30]. Thus, daily administration of IL-15 exhibited increased *in vivo* toxicity compared to intermittent administration. Daily administration also led to an expansion of effector, but not memory, T cells, while intermittent dosing enhanced memory T cells [44].

IL-15 has also been evaluated for intravenous delivery. Intravenous delivery was associated with higher toxicity compared to subcutaneous delivery [30,45]. When administered at a high dose and frequency to rhesus macaques (RMs) (50 mcg intravenously daily for 12 days), the primary side effect was transient grade 3/4 neutropenia that recovered after ending IL-15 treatment [42,43]. Surprisingly, there was no relationship between the dose (10 and 50 μg delivered intravenously) and immunomodulatory effects of IL-15 [43]. Taken together, these data indicate that there is a delicate balance between efficacy and *in vivo* toxicity of IL-15 treatment that is related to route, dose, and timing of administration.

## 4. Effects of IL-15 on T Cells in Healthy and HIV/SIV-Infected Individuals

IL-15 can promote the function and survival of virus-specific CD8^+^ T cells both *in vitro* and *in vivo*, which is one reason why many have considered IL-15 as part of a strategy for a functional cure for HIV. Antigen-specific CD8^+^ T cells are critical players in controlling HIV infection. Modulating them for increased function and longevity could be a crucial component of an HIV cure strategy [46,47,48]. *In vitro*, spontaneous apoptosis and CD95/Fas-induced apoptosis of HIV-specific CD8^+^ T cells were inhibited when PBMCs (peripheral blood mononuclear cells) from HIV^+^ individuals were treated with IL-15, and the long-term survival of CD8^+^ T cells, notably HIV-specific CD8^+^ T cells, was increased [49]. IL-15 treatment *in vitro* also increased antigen-specific activation of HIV-specific CD8^+^ T cells as measured by increased CD69 expression [49]. *Ex vivo*, treatment of PBMC from HIV^+^ donors with IL-15 did not affect the frequency of IFN-γ-producing bulk CD8^+^ T cells, but IL-15 treatment did increase the frequency of IFN-γ-producing HIV-specific CD8^+^ T cells and the direct *ex vivo* cytotoxicity of HIV-specific cells [49]. Additionally, in a separate study, treatment of peripheral CD4^+^ and CD8^+^ effector memory T (T_EM_) cells isolated from HIV^+^ donors with IL-15 *ex vivo* enhanced cytokine production and prevented apoptosis, and unpublished observations by Meuller and Katsikis suggest that IL-15 treatment may prevent apoptosis of HIV-specific CD8^+^ T cells as well [50].

The *in vivo* effects of IL-15 treatment on SIV-specific CD8^+^ T cells appear more nuanced than the *in vitro* results. *In vivo* IL-15 treatment had no effect on the number of IFN-γ-secreting SIV-specific CD8^+^ T cells in SIV^+^ macaques infected either intrarectally or intravaginally, despite increasing the bulk CD8^+^ T_EM_ and NK cell populations [51]. The failure of IL-15 treatment to boost IFN-γ-secreting SIV-specific CD8^+^ T cells in the PBMCs of animals may result from either the dose or duration of treatment. The CD8^+^ T_EM_ cells induced by IL-15 may also have functions aside from IFN-γ secretion, which cannot be detected by IFN-γ ELISpot. Repeating these studies with activation-induced marker (AIM) assays may potentially detect T cell activation that was undetected by IFN-γ ELISpot. Finally, in ART-treated, chronically (>9 months) SIV^+^ macaques, SIV-specific CD8^+^ T cells transiently proliferated in response to IL-15 treatment, while SIV-specific CD4^+^ T cells did not [52]. These studies provide evidence that IL-15 therapy may hold potential in CD8^+^ T cell-based HIV/SIV therapies but may be a less effective agent in CD4^+^ T cell-based therapies.

Treatment of animals with IL-15 *in vivo* is also complex regarding how CD4^+^ and CD8^+^ T cells respond. *In vivo*, the frequency of CD8^+^ T cells dramatically increases in the peripheral blood of SIV-naïve and SIV^+^ macaques treated with IL-15, while CD4^+^ T cells typically increase in number only moderately [43,45,51,52]. However, there are conflicting data regarding whether certain phenotypes of CD4^+^ T cells expand and in what circumstances (SIV^+^ versus SIV-naïve, ART-treated versus ART-naïve, etc.) this occurs. In SIV^+^ ART-naïve cynomolgus macaques infected intrarectally or intravaginally, IL-15 treatment did not increase proliferating (Ki-67^+^) CD4^+^ T cells, but did increase proliferating (Ki-67^+^) CD8^+^ T cells in the peripheral blood, suggesting expansion of CD8^+^ T cells [51]. These proliferating CD8^+^ T cells are generally T_EM_ and/or central memory T (T_CM_) cells [31,42,51,53]. There is evidence that IL-15 treatment dramatically increases the number of CD8^+^ T_EM_ in the peripheral blood of SIV-naïve and SIV^+^ ART-naïve macaques when compared to CD8^+^ T_CM_ cells, indicating a preferential expansion of the CD8^+^ T_EM_ subset during IL-15 treatment [45,51]. On the other hand, increased production of CD4^+^ T_EM_ cells have also been shown following IL-15 treatment in SIV^+^ animals infected intravenously that are virally suppressed on ART [54]. Indeed, IL-15 treatment of ART-suppressed SIV^+^ RMs infected intravenously increased expression of Ki-67 on both CD4^+^ and CD8^+^ T_EM_ [55]. These data suggest that IL-15-dependent expansion of memory CD4^+^ and CD8^+^ T cell populations may be dependent on the extent of ongoing SIV-dependent immune dysregulation. The degree of *in vivo* CD4^+^ and CD8^+^ T cell sensitivity to IL-15 appears to be dependent upon whether SIV^+^ animals are receiving ART. In ART-naïve SIV^+^ macaques infected intravenously, IL-15 administration has little to no effect on increasing the proliferation of CD4^+^ T_EM_ or transitional memory T (T_TM_) cells and exhibits only modest effects on CD8^+^ T_EM_ [54]. This is likely due to an increased (2- to 4-fold higher) steady-state proliferation of memory CD4^+^ and CD8^+^ T cells in untreated SIV^+^ animals compared to SIV-naïve animals [54,56]. However, IL-15 sensitivity was restored upon virus suppression with ART; steady-state CD4^+^ and CD8^+^ T_EM_ proliferation was decreased, and T cells then proliferated when animals received IL-15 [54]. This suggests that aberrant immune activation induced by uncontrolled SIV infection may reduce immune responsiveness to IL-15 treatment, while protected immune systems such as in SIV^+^ animals virally suppressed on ART may be required for IL-15 responsiveness.

Finally, IL-15 dramatically alters the *in vivo* migration and trafficking of CD4^+^ and CD8^+^ T cells in SIV-naïve and SIV^+^ macaques [57]. While IL-15 can induce proliferation, the cells may traffic to sites aside from the peripheral blood, complicating the analysis of proliferation versus migration. Both naïve and memory CD4^+^ and CD8^+^ T cells traffic from the blood vessels to peripheral tissues following IL-15 treatment of SIV-naïve RMs [43]. In ART-treated SIV^+^ RMs infected intravenously, delivery of IL-15 induced CD4^+^ and CD8^+^ T cell emigration to extra-lymphoid effector sites such as the bronchoalveolar space [54]. This IL-15-directed localization of memory T cells to mucosal sites, where HIV/SIV replicates, represents a potential immunotherapeutic target in the context of HIV.

## 5. Effects of IL-15 on NK Cells in Healthy and HIV/SIV-Infected Individuals

IL-15 is critical for NK cell development, survival, and homeostasis, demonstrated by near-complete NK cell depletion following IL-15 blockade in SIV-naïve RMs [58]. Conversely, treatment with IL-15 can enhance the number, frequency, and function of NK cells in SIV-naïve RMs [58], and induces NK cell activation and expansion in both SIV-naïve and SIV^+^ RMs [59]. Similar to the effects seen in T cells, both daily and intermittent administration of IL-15 via both intravenous and subcutaneous routes significantly increase the number of circulating NK cells in rhesus, cynomolgus, and pigtailed macaques [30,42,45,51].

Though it is unclear whether ART affects the sensitivity of NK cells to IL-15 *in vivo*, *in vitro* treatment of PBMC from ART-treated HIV^+^ individuals with IL-15 restored NK cell-mediated cytotoxic function and IL-12 production that were deficient due to HIV infection [60]. In *ex vivo* autologous HIV replication systems consisting of NK cells, CD4^+^ T cells, and reservoir virus all isolated from the same donor, IL-15 treatment improved many NK cell functions, including NK cell receptor expression, antibody-dependent cellular cytotoxicity (ADCC), IFN-ɣ production, and cytotoxicity by degranulation [61]. IL-15 stimulation also mediated the clearance of HIV-1-infected cells by NK cells following latency reversal with the histone deacetylase (HDAC) inhibitor Vorinostat in an *ex vivo* viral outgrowth assay [61]. Thus, IL-15 exhibits marked effects on enhancing NK cell number and function both *in vivo* and *in vitro*.

## 6. Effects of IL-15 on Chronic Plasma HIV/SIV Viremia

We do not currently know whether IL-15 treatment favors virus replication, perhaps out of latency, or immunological suppression, via enhanced T cell function. By measuring the host viral load, we can assess the impact of both virus activation and virus suppression within a single host. IL-15 treatment does not appear to increase the long-term HIV/SIV plasma viral burden and may contribute to decreasing viral burden. No statistically significant differences in viral load changes from baseline were observed between chronically infected SIV^+^ macaques treated with IL-15 and untreated controls [51]. Yet, in ART-naïve chronic SIV^+^ macaques, IL-15 treatment led to a modest, temporary viral load decrease [54]. Together, these studies suggest IL-15 administration during chronic SIV infection of ART-treated animals may be therapeutically beneficial in enhancing antiviral immune responses without significantly increasing the plasma viral burden.

The relationship between natural plasma IL-15 levels and viral load may also provide some insight into whether the presence of this cytokine can provide a beneficial effect on antiviral immunity. In HIV^+^ humans, one study found the natural plasma IL-15 level in chronically infected HIV^+^ patients during analytical treatment interruption (ATI) indicated that patients with higher IL-15 levels were able to control viral replication even during ART interruption, compared to patients with lower IL-15 levels, suggesting a potentially beneficial role for IL-15 in chronic infection [62]. The impacts of IL-15 expression and administration during acute SIV will be addressed later in this review.

## 7. IL-15 in Combination with Vaccination

IL-15 and IL-15 agonist complexes have also been evaluated for their ability to enhance vaccine-elicited immunity in many diseases, such as tetanus and influenza [44]. Indeed, IL-15 may have the potential to act as an adjuvant with additional therapies to maximize immune responsiveness in the context of HIV, particularly under ART control. IL-15 has been assessed as an HIV DNA vaccine adjuvant in four clinical trials (NCT00775424, NCT00115960, NCT00528489, NCT00195312). DNA vaccinations by themselves are typically poor inducers of antigen-specific immunity, so they have been combined with IL-15 in an attempt to maximize vaccine potency and vaccine-induced cellular immunity [63,64].

Studies in mice have investigated whether IL-15 can boost vaccine-elicited immune responses. To characterize the requirement of IL-15 for the development and maintenance of a vaccine-induced cytotoxic T cell response, Li and colleagues evaluated the immunogenicity of SIV DNA-based vaccines in IL-15 knockout mice [65]. While vaccination with SIV Gag DNA led to induction of antigen-specific CD4^+^ and CD8^+^ T cells in the absence of IL-15, the absolute number of antigen-specific CD8^+^ T cells was lower in IL-15 knockout mice than in their WT counterparts [65]. This indicates that IL-15 is needed to maximize the number of antigen-specific CD8^+^ T cells generated by DNA vaccination [65]. Additionally, they found that while IL-15 was not essential for the long-term maintenance of vaccine-induced antigen-specific memory T cells, it was essential for granzyme B production and, therefore, cytolytic capacity of cytotoxic CD8^+^ T cells elicited by vaccination. Oh and colleagues found that mice immunized with vaccinia virus vectors expressing IL-15 and HIV gp160 (the glycoprotein encoded by the HIV Env gene) developed strong and long-lasting antibody-mediated immunity, a short-term cytotoxic T cell response against HIV gp120, and robust long-term CD8^+^ T cell-mediated immunity [66]. Although this was not a DNA vaccine, the results are consistent with those in studies of DNA vaccines, suggesting that IL-15 is beneficial, but not required, for the induction of long-lived, vaccine-induced, antigen-specific memory CD8^+^ T cells in mice.

IL-15 has been included as part of therapeutic vaccination regimens to SIV/HIV because of the potential for inducing and maintaining cellular immune responses, and the results are mixed. When ART-suppressed, SIV^+^ RMs infected intravenously were therapeutically vaccinated with SIV-DNA with or without IL-12, boosting with SIV-DNA and plasmid IL-15 (compared to boosting with nothing or SIV-DNA alone without IL-12) enhanced CD4^+^ and CD8^+^ T_EM_ cytokine production and prompted dual-functionality (production of both IFN-ɣ and TNF-α) of SIV-specific memory T cells [63]. This suggests that delivery of plasmid IL-15 can enhance immune induction by therapeutic DNA vaccines in the context of ART-suppressed SIV infection. In contrast to the beneficial effect of IL-15 on DNA vaccines, ART-suppressed, SIV^+^ RMs infected intravenously, immunized with the live poxvirus vaccine, ALVAC-SIV-gpe (expressing SIV gag, pol, and env), exhibited an increased viral set point during ATI when treated with IL-15, when compared to RMs that were vaccinated but did not receive IL-15 [55]. Additionally, IL-15 did not significantly augment the vaccine-induced Gag-specific immune response. These studies suggest that IL-15 may enhance the immunogenicity of therapeutic DNA vaccination, but may not enhance the immunogenicity of therapeutic vaccination utilizing a live viral vector, and indeed may negatively impact viral set point in the context of live viral vector vaccine regimens. Additional studies will be required to determine whether it is possible for IL-15 to enhance the immunogenicity of vaccinations comprised of viral vectors, and whether IL-15 consistently increases the viral set point following ATI or whether this was a consequence of IL-15 treatment combined with this specific vaccine regimen.

IL-15 has also been studied in the context of prophylactic HIV/SIV vaccination regimens to elicit antiviral T cells primed to suppress viremia upon HIV/SIV exposure, but these results have also been mixed. Boyer and colleagues vaccinated naïve animals with a DNA vaccine encoding SIVgag, SIVpol, and HIV-1env alone or in combination with an optimized plasmid encoding macaque IL-15 [67]. The animals vaccinated with this regimen were still susceptible to intravenous infection with SHIV89.6P, but animals that received plasmid IL-15 controlled infection more quickly than the group treated with DNA vaccine alone. Furthermore, all the vaccinated animals in this study experienced a strong induction of vaccine-specific, IFN-ɣ-producing CD4^+^ and CD8^+^ effector T cells. In contrast, two studies evaluated HIV-1 DNA vaccines with plasmid IL-15 in HIV-naïve humans. The IL-15 was well tolerated, but did not appear to augment immunogenicity of the prophylactic vaccines [68,69]. Whether a failure for IL-15 to boost vaccine immunogenicity was a consequence of the selected vaccine or a failure of IL-15 to boost immune responses is unclear. More studies are needed to define whether IL-15 can serve as a prophylactic vaccine adjuvant to enhance and maintain elicited cellular immunity.

## 8. IL-15, Naturally Produced or Delivered, in ART-Suppressed, Untreated, Acute, and Chronic HIV/SIV

Throughout HIV infection, levels of IL-15 naturally produced by the body transiently peak during acute infection and wane during chronic infection [70]. IL-15 levels elevate to a peak at day 10 post-infection and subsequently decline by day 14 post-infection [71]. However, there is conflicting evidence as to whether chronic IL-15 levels are elevated in HIV-infected individuals and how IL-15 levels are affected by ART treatment. Swaminathan and colleagues found that HIV^+^ individuals with high viral loads (>100,000 copies/mL) had significantly higher plasma IL-15 levels than HIV^+^ individuals with low viral loads (<50 copies/mL) or those who were HIV-naïve, though it is unclear whether high levels of viremia promoted cytokine signaling or whether cytokine production was a driving factor in viral replication [72]. Additionally, IL-15 levels in the lymph nodes of ART-naïve HIV^+^ individuals were nearly 2-fold higher than those in the lymph nodes of ART-suppressed HIV^+^ individuals or healthy controls, and these elevated LN IL-15 levels were a predictor of disease progression [73]. Conversely, an evaluation by Boulassel and colleagues revealed no significant differences in plasma IL-15 levels between healthy subjects and HIV^+^ individuals with high, low, and undetectable viremia [74]. Thus, while much of the available evidence suggests that IL-15 levels do significantly increase in acute HIV and subside during chronic infection [70], it is unclear whether IL-15 levels subside to levels similar to those of HIV-naïve individuals, or remain elevated, albeit less elevated than in acute infection.

How the levels of IL-15 during acute and chronic HIV infection impact disease course is not fully understood. Indeed, chronic immune activation is associated with accelerated SIV pathogenesis [75]. Because IL-15 is one of the first cytokines to be produced in response to acute HIV infection [70] and IL-15 can be induced by viral infections [20], it is conceivable that viral replication can promote IL-15 signaling, thereby enabling chronic immune activation and disease progression. Thus, additional studies that include ART suppression would be beneficial to determine whether suppression of virus replication with ART affects IL-15 levels and chronic disease pathogenesis. Studies with and without ART suppression will likely be essential in further defining the dynamics between HIV/SIV replication or lack thereof, IL-15 levels, and disease progression.

IL-15 treatment in acute HIV/SIV and at the time of ATI may contribute to enhancing SIV pathogenesis. Acute HIV/SIV infection is characterized by immune activation [76,77], and studies have shown that IL-15 can enhance HIV replication in addition to antigen-induced lymphocyte proliferation [78,79]. Macaques treated with IL-15 during acute SIV exhibited a 1000-fold increase in viral load set point in chronic infection and accelerated progression to AIDS [80,81]. Thus, it appears that IL-15 treatment during acute infection would not be advantageous in the context of HIV/SIV. Additionally, when chronically infected RMs were simultaneously initiated on a therapeutic regimen consisting of both ART and IL-15, IL-15 delayed viral suppression, but did not abrogate it [52]. The macaques that received IL-15 with ART also lost CD4^+^ T cells faster than macaques receiving ART alone upon ATI, suggesting that the timing of ART initiation may also be a poor time to administer IL-15 in the context of HIV/SIV.

Elevated IL-15 in acute HIV regulates the susceptibility of CD4^+^ T cells to infection, thus playing a crucial role in viral reservoir establishment [82]. In RMs, natural plasma IL-15 levels during acute SIV were significantly positively correlated with viral infection of CD4^+^ T cells, and this was attributed specifically to an increase in CD4 on the surface of memory CD4^+^ T cells between day 7 and 10 in acute SIV [71]. IL-15 may drive higher viremia in acute SIV because it increases the susceptibility of target CD4^+^ T cells to HIV/SIV infection [71,72,82]. The findings of these studies indicate that high levels of IL-15 or IL-15-based therapy in acute HIV/SIV infection may promote reservoir establishment, increase viral load and post-peak viral load set point, and accelerate disease progression. However, it is important to note that Okoye et al. found that target CD4^+^ T cell activation and expansion is not the only factor that influences viral load. Contradicting the prevailing hypothesis that CD4^+^ T cell activation and proliferation enhance HIV/SIV pathogenesis by expanding CD4^+^ target cell populations, the initial establishment of post-peak viral load set points in acute SIV is more closely associated with the functional ability of CD8^+^ T cells [81]. When RMs underwent IL-15 blockade during primary or chronic SIV infection, SIV replication and disease progression were comparable to control groups, suggesting IL-15 is not solely responsible for increased viremia or progression to AIDS [83]. Thus, this is an area where more research is needed to elucidate the extent of the contributions of each cell population to HIV/SIV pathogenesis. It is likely a combination of the availability of CD4^+^ target cells and their susceptibility to infection, as well as the availability and functional capacity of SIV-specific CD8^+^ T cells, that determines viral load, viral load set point, and disease progression.

Activation of the “right” components of the immune system at the “right” time may be a key component of immune-based HIV remission strategies. Although IL-15 may have detrimental effects during acute HIV infection, the beneficial effects of IL-15 on CD4^+^ T cells, CD8^+^ T cells, and NK cells discussed earlier in this review remain valid. The ability of IL-15 to modulate the trafficking and function of these cells, in addition to its potentially beneficial effects on viral load reduction, support a therapeutic role for IL-15 in chronic infection. It may be important to reserve IL-15 therapies until chronic infection to maximize beneficial effects on T_EM_ cells and viral control while minimizing potential risk to the viral reservoir, viral load set point, and immune exhaustion. It is possible that the ideal time to deliver IL-15 would be after ART initiation or perhaps following ATI in order to allow for necessary immune activation while preventing aberrant activation or exhaustion. Therefore, the timing of IL-15 blockade and administration appears to be a key factor for immune stimulation at the proper time to control SIV while avoiding chronic immune activation, exhaustion, and SIV progression.

Finally, the location of IL-15 signaling is likely as important as the timing of IL-15 production or delivery. One distinguishing feature of models of SIV non-progression (African green monkeys, AGMs) compared with SIV progression (RMs) is IL-15-mediated control of SIV replication in the LNs by NK cells [83]. Non-progression is characterized by relatively stable NK cell frequencies in the LNs while progression is characterized by a steady decline in LN NK cell frequencies. Elevated IL-15 in the LN B cell follicles produced by follicular dendritic cells is associated with NK cell antiviral activity and localization in the LNs [84]. Indeed, NK depletion in SIV^+^ AGMs via treatment with an anti-IL-15 monoclonal antibody led to enhanced viral replication in the LN B cell follicles and T cell zones [85]. While the factors contributing to enhanced IL-15 in the LNs of non-progression models compared to progression models are not well understood, this suggests an important role for IL-15 in the natural control of SIV replication in a natural host model of SIV non-progression, and thus represents a potential therapeutic target in developing a functional cure [86].

## 9. Benefits of Using an IL-15 Superagonist, Such as N803, over Free IL-15

The use of free IL-15 as an immunotherapeutic agent may be limited because the efficacy of IL-15 signaling is dependent on the coupling of IL-15 to IL-15Rα [31]. The utility of delivering free IL-15 to animals may be dependent on the availability of free IL-15Rα, as the coadministration of IL-15 and IL-15Rα enhanced IL-15 activity ~50-fold compared to IL-15 alone [31,87]. The half-life of free IL-15 is less than one hour (it is quickly degraded and is also under tight transcriptional and translational regulation), limiting its *in vivo* bioactivity and, by extension, limiting its therapeutic capacity [88]. For this reason, different mechanisms by which IL-15 signaling may be boosted by increasing the efficacy and half-life both *in vitro* and *in vivo* have been explored. One way to improve the stability and extend the half-life of IL-15 is to produce an IL-15:IL-15Rα agonist complex [31,89]. Various IL-15 agonist complexes have been developed to maximize the biological activity and half-life of IL-15 to improve its immunotherapeutic potential for the treatment of chronic diseases such as cancer and HIV. The structure and immunobiology of many of these complexes have been reviewed by Guo and colleagues [53]. The focus of this review will be on N-803, formerly Alt-803, because it is the primary IL-15 superagonist complex being studied in clinical trials as an HIV therapeutic. N-803 has also been evaluated for its ability to reverse latency while concurrently boosting cellular immune functions [90,91,92,93], highlighting its relevance to HIV cure strategy development.

N-803 was developed by Altor Biosciences to overcome some of the drawbacks observed with IL-15 treatment, such as its short half-life and limited efficacy. N-803 consists of two IL-15 molecules with an asparagine to aspartic acid change at amino acid 72 (N72D), each bound to the sushi domain of an IL-15Rα molecule, both fused to the Fc region of human IgG1 [94]. The N72D change in N-803 confers a 5-fold increase in biological activity compared to free IL-15, and the IgG1 Fc confers stability and increases the half-life of the complex [94]. With both of these changes, N-803 exhibits 25-fold higher biological activity and 35-fold longer serum half-life than free IL-15 [94,95,96,97]. N-803 exhibits greater tissue distribution and longer persistence in lymph nodes compared to IL-15 as well [98]. While other IL-15 agonist complexes have been developed, they have not shown significant improvement upon the efficacy of N-803, so they have not been pursued beyond basic *in vitro* assessment and we will not discuss them here [99].

Subcutaneous and intravenous N-803 treatment is safe and well-tolerated in mice, macaques, and humans *in vivo* [96,97,98,100,101,102]. Intravenous N-803 administration is associated with 100-fold higher serum concentrations compared to subcutaneous administration, but subcutaneous N-803 administration exhibits greater biodistribution to lymphoid organs [96,103]. The proliferation and activation of CD8^+^ T cells and NK cells, however, was comparable between intravenous and subcutaneous administration [103]. Because N-803 is primarily being given to humans subcutaneously in clinical trials studying N-803 in both healthy individuals and those with cancer, this route of administration may be the most clinically relevant (NCT02523469, NCT03381586, NCT03054909, NCT03022825). The major side effect in cynomolgus macaques is dose-dependent inappetence, and in humans, the major side effects are injection site reactions, fatigue, and nausea [96,98,104]. Intravenous N-803 is well tolerated up to 0.1 mg/kg weekly in mice and cynomolgus macaques, and has been evaluated up to 0.01 mg/kg weekly in humans with no dose-limiting toxicities defined [98,102]. Because N-803 exhibits enhanced pharmacokinetics and biodistribution compared to free IL-15 *in vivo*, N-803 may be a preferred immunotherapeutic intervention.

## 10. Effects of N-803 on T Cells and NK Cells

N-803 can enhance the proliferation, activation, and function of CD8^+^ T cells and NK cells, both *in vitro* and *in vivo* [90,98,105]. In SIV-naïve and SIV^+^ ART-naïve RMs, N-803 treatment transiently elevated the total CD4^+^ T_EM_, CD8^+^ T_EM_, CD4^+^ T_CM_, CD8^+^ T_CM_, and NK cell populations in peripheral blood [100,101]. Subcutaneous administration of N-803 induced activation and proliferation of CD8^+^ T cells and NK cells in SHIV-infected ART-suppressed RMs, further suggesting that N-803 may be an effective adjuvant to boost the cellular immune response in the context of SIV with or without virus suppression under ART [91].

Additionally, N-803 treatment induced trafficking of both NK cells and SIV-specific CD8^+^ T cells from the peripheral blood to the B cell follicles of the lymph nodes of SIV^+^ ART-naïve RMs [101]. This N-803-mediated trafficking to the lymph nodes also occurred in ART-suppressed, SHIV-infected (intravenous) macaques, suggesting that N-803 can induce antigen-specific CD8^+^ T cells and NK cells to traffic to the lymph nodes in both treated and untreated HIV/SIV infection [91]. This lymph node trafficking is critical because the B cell follicles of lymph nodes represent a major location of HIV/SIV reservoir establishment in CD4^+^ T cells [106,107]. Yet, HIV- and SIV-specific CD8^+^ T cells are largely excluded from the follicular space [4,108,109]. Because N-803 improves the trafficking of SIV-specific CD8^+^ T cells to the B cell follicles of the lymph nodes, this may help localize antiviral CD8^+^ T cells to the site of virus replication where they can encounter infected cells, thereby overcoming a major barrier to achieving immune control of HIV or SIV replication [84]. However, the studies summarized here were all completed in SIV^+^ or SHIV^+^ macaques, so whether treatment with N-803 would engender the same effects in HIV^+^ humans remains unclear.

## 11. Effects of N-803 on SIV Replication

The impact of N803 on SIV viral load is nuanced. N-803 treatment in ART-naïve, SIV^+^ RMs that were previously elite controllers led to a transient decrease in viral loads in one study by Ellis-Connell et al., but no consistent viral load changes were observed in a similar study by Webb et al. [100,101]. However, in this study by Webb and colleagues, the one elite controller animal with detectable viremia experienced a ~1 log viral load decrease following treatment [101]. These studies suggest N-803 may mediate viral load reductions in viremic elite controllers. More studies are necessary to elucidate the potential role of N-803 in suppressing viral replication and why this occurs in some cases, but not in others.

## 12. N-803 as a Latency-Reversing Agent

One approach of interest for an HIV cure is the “shock and kill” strategy. This strategy first requires a latency-reversing agent (LRA) to “shock” the virus out of latency by inducing HIV antigen expression in latently infected cells. Immune cells are subsequently required to “kill” those infected cells, thereby reducing the size of the viral reservoir [48,110,111,112]. N-803 is of interest in latency reversal because it can simultaneously induce virus replication while modulating NK cell and T cell function and proliferation [112]. N-803 may disrupt the latent reservoir, prime latently infected targets for recognition by CD8^+^ T cells, and subsequently increase CD8^+^ T cell numbers and effector function, thereby mediating both the “shock” and “kill” aspects of this strategy. One nuance to this approach, particularly in CD8^+^ T cell-based shock and kill strategies, is that an ideal LRA must produce detectable amounts of antigen recognizable to CD8^+^ T cells but need not induce virion production [90]. Although CD8^+^ T cells can detect minimal amounts of antigen on target cells, they may be unable to detect cell-associated HIV RNA alone; LRA-mediated peptide production and presentation may be necessary for T cell recognition. However, virion production may lead to bystander T cell activation and the release of infectious virions, which may result in complications in HIV^+^ individuals on or off ART [113]. For these reasons, the ideal LRA would induce small amounts of antigen presentation on latently infected cells while not inducing viral assembly or budding of virions. One strategy to accomplish this goal is to attempt to reverse latency under the control of ART, but many LRAs being studied in clinical trials are insufficient to provide a robust enough “shock” to induce reactivation of the latent viral reservoir in HIV^+^ patients virally suppressed on ART [114,115,116,117,118].

*In vitro*, N-803 can both reverse HIV latency and improve CD8^+^ T cell recognition of infected cells [90,92]. When latently infected PBMCs from HIV-infected donors were depleted of CD8^+^ T cells and activated cells (expressing CD69 or HLA-DR), N-803 treatment reversed HIV latency and induced detectable antigen expression, enhancing HIV-specific CD8^+^ T-cell recognition *ex vivo* [90]. In an *in vitro* model of HIV latency, N-803 reactivated expression of HIV Gag in latently infected CD4^+^ T cell monocultures absent of HIV-specific CD8^+^ T cells [92]. Further, co-culture with activated CD8^+^ T cells in this *in vitro* latency system significantly inhibited N-803-mediated reactivation, suggesting that N-803-mediated latency reversal is suppressed by CD8^+^ T cell activity *in vitro*.

*In vivo*, N-803 has exhibited latency-reversing potential, but only in the absence of CD8^+^ T cells. When N-803 was administered to ART-suppressed, SHIV-infected (intravenous) RMs, there were transient episodes of SHIV RNA detected in the plasma [91]. There was no significant difference in the area under the curve of these viral load “blips” between animals that did or did not receive N-803 and there was no difference in the size of the cell-associated viral DNA reservoir following N-803 administration in animals that received N-803, suggesting that N-803 alone is not sufficient to reverse latency in ART-suppressed, SHIV-infected RMs [91]. However, N-803 treatment induced virus production when ART-suppressed SIV^+^ RMs infected intravenously underwent CD8α depletion or SHIV-infected, ART-suppressed RMs infected intrarectally underwent CD8α or CD8β depletion at the time of N-803 treatment [92,93]. Because CD8^+^ lymphocytes are critical for virus suppression both in untreated SIV and in SIV under the control of ART [119], N-803 may still reactivate SIV from latency, but uncontrolled virus replication may be reduced by the presence of these CD8^+^ T cells. While treatment of SIV- (intravenous) or SHIV-infected (intrarectal), ART-suppressed RMs with N-803 during CD8α depletion effectively induced latent reactivation, this did not specifically implicate CD8^+^ T cells as the cells preventing latency reactivation as CD8α depletion targets other cells that express CD8α, such as NK cells, in addition to CD8^+^ T cells [92]. Treatment of SHIV-infected (intrarectal), ART-suppressed RMs with N-803 during CD8β depletion, on the other hand, also effectively induced reactivation of the latent viral reservoir, despite suboptimal depletion of CD8^+^ T cells [93]. This indicates that concurrent depletion of CD8^+^ T cells with N803 treatment reveals the latency-reversing potential of N-803. It is clear that more work is required to define the necessary parameters of N-803 as an LRA. Still, the results of these studies show N-803 as a promising combination candidate agent for latency reversal and immune activation.

## 13. Concluding Remarks

IL-15-based therapies hold promise as an immunostimulatory component of an HIV therapeutic regimen or potential cure strategy. While antiretrovirals are very effective at controlling HIV infection, this control is lost upon ART cessation. The common ɣ chain cytokine IL-15 exhibits numerous functions in mediating the adaptive cellular immune response without the considerable toxicity of IL-2, making it an attractive immunotherapeutic target for promoting long term viral control of HIV/SIV infection via CD8^+^ T cells. IL-15 administration has been shown to be safe in macaques and humans, causing only minimal and reversible toxicity. IL-15 treatment enhances the proliferation, survival, and tissue migration of NK cells and CD8^+^ T cells, specifically memory and antigen-specific CD8^+^ T cells. While IL-15 does not appear to increase viral load, it may contribute to reducing viral load, and more research is necessary to elucidate the roles of both naturally produced IL-15 as well as treatment with IL-15 in HIV/SIV disease progression. IL-15 has also been evaluated in combination with vaccination to enhance HIV/SIV-specific immunity, and IL-15 has been shown in most circumstances to augment the number, function, maintenance, and longevity of vaccine-elicited cellular immune responses. However, there have been cases in which using IL-15 as a vaccine adjuvant resulted in an increased viral setpoint and no augmentation of the antigen-specific cellular immune responses, though it remains unclear why IL-15 was a beneficial addition to most vaccine regimens, but not for all. Finally, while IL-15 administration during acute HIV/SIV or at the time of ATI may cause aberrant immune activation and exhaustion (and, by extension, disease progression), IL-15 administration during chronic infection of ART-treated animals appears to hold therapeutic potential to augment antiviral immune responses without significantly increasing viremia.

N-803 has a significantly longer half-life and greater biological activity and efficacy compared to IL-15. Similar to IL-15, N-803 has been shown to enhance the activation, proliferation, function, and trafficking of CD8^+^ T cells and NK cells to mucosal sites and lymph nodes, which are particularly beneficial characteristics in the context of HIV/SIV infection. Limited data also suggests N-803 may have the potential to mediate viral load reductions in SIV^+^ viremic elite controller macaques, but more research is required to define the parameters of N-803-mediated viral load decreases. Lastly, N-803 has been evaluated for its ability to reverse latency and enhance CD8^+^ T cell recognition and killing of infected target cells to both “shock” and “kill” the SIV reservoir, though recent studies suggest that N-803 alone may be insufficient to accomplish this *in vivo*. Many of these therapeutic interventions (e.g., IL-15, N-803, vaccination, ART) have been evaluated in combination, and while IL-15 and N-803 have demonstrated potential as HIV immunotherapeutics, the ideal timing and combination of IL-15/N-803 with additional interventions remains to be defined. Any HIV cure-strategy will undoubtedly require a combination of the various approaches to achieve a functional cure, and IL-15/N-803 intervention may be one such component.

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
