# Peer review of "Therapeutic Potential of IL-15 and N-803 in HIV/SIV Infection"

_viruses, 2021, doi:10.3390/v13091750_

Round 1

Reviewer 1 Report

The manuscript provides a comprehensive review of the role of IL-15 and N-803, a IL-15 superagonist, in HIV/SIV infection. The information is well organized. The followings are few comments.

  1. Line 39, early or delay ART treatment does not prevent immune activation or exhaustion in HIV+ patients. The statement regarding that ART help prevent immune exhaustion should be revised.
  2. Line 71, delete “l” after cytotoxicity.
  3. Lines 99 and 324. Please specify the levels of IL-15 when authors refer “high” levels of IL-15.
  4. In the section 3, it would be helpful if authors describe the parameters for assessing cytotoxicity.
  5. In the section 8, please describe IL-15 levels in SIV progression (RMs) and non-progression (African green monkey) models.
  6. Authors described the route of administration of IL-15 and N-803 in macaque studies. However, it will be helpful if authors describe the route of viral infection in the animal models. Different viral doses and routes may lead to different outcomes.
  7. It would be helpful if authors described the effect of IL-15 and N-803 on HIV infection of PBMCs or other primary cells.

Reviewer 2 Report

This manuscript is very well written and organized. Reviewing the current literature on the role of IL-15 in HIV-SIV infection is an important contribution for the field, especially in light of recent works describing the IL-15 super-agonist N-803 as a potential LRA. I suggest the authors to discuss the role of type I IFN, which is fundamental during viral infections, in inducing expression and production of IL-15. Mattei F, Schiavoni G, Belardelli F, Tough DF (2001) IL-15 is expressed by dendritic cells in response to type I IFN, double-stranded RNA, or lipopolysaccharide and promotes dendritic cell activation. J Immunol 167:1179–1187. Hansen, M.L. et al. IFN-alpha primes T- and NK-cells for IL-15-mediated signaling and cytotoxicity. Mol. Immunol. 48, 2087–2093 (2011). In addition I would change the sentence line 25-26. While it is true that CD4 T cells are the main target of HIV infection and contributors to the viral reservoir, other immune cells are involved (monocytes and macrophages) and it should be acknowledged.
